# Phenol-soluble modulin α and β display divergent roles in mice with staphylococcal septic arthritis

Zhicheng Hu [1,2], Pradeep Kumar Kopparapu [1,7], Patrick Ebner[3,7], Majd Mohammad [1], Simon Lind[1], Anders Jarneborn[1,4], Claes Dahlgren[1], Michelle Schultz [1], Meghshree Deshmukh[1], Rille Pullerits [1,5], Mulugeta Nega[3], Minh-Thu Nguyen [6], Ying Fei[2], Huamei Forsman[1], Friedrich Götz [3] & Tao Jin [1,4 ✉]

Phenol-soluble modulin α (PSMα) is identified as potent virulence factors in *Staphylococcus aureus* (*S. aureus*) infections. Very little is known about the role of PSMβ which belongs to the same toxin family. Here we compared the role of PSMs in *S. aureus*-induced septic arthritis in a murine model using three isogenic *S. aureus* strains differing in the expression of PSMs (Newman, Δ*psmα*, and Δ*psmβ*). The effects of PSMs on neutrophil NADPH-oxidase activity were determined in vitro. We show that the PSMα activates neutrophils via the formyl peptide receptor (FPR) 2 and reduces their NADPH-oxidase activity in response to the phorbol ester PMA. Despite being a poor neutrophil activator, PSMβ has the ability to reduce the neutrophil activating effect of PSMα and to partly reverse the effect of PSMα on the neutrophil response to PMA. Mice infected with *S. aureus* lacking PSMα had better weight development and lower bacterial burden in the kidneys compared to mice infected with the parental strain, whereas mice infected with bacteria lacking PSMβ strain developed more severe septic arthritis accompanied with higher IL-6 and KC. We conclude that PSMα and PSMβ play distinct roles in septic arthritis: PSMα aggravates systemic infection, whereas PSMβ protects arthritis development.

[1] Department of Rheumatology and Inflammation Research, Institute of Medicine, Sahlgrenska Academy, University of Gothenburg, Gothenburg, Sweden. [2] Center for Clinical Laboratories, the Affiliated Hospital of Guizhou Medical University, Guiyang, China. [3] Department of Microbial Genetics, Interfaculty Institute of Microbiology and Infection Medicine Tübingen (IMIT), University of Tübingen, Tübingen, Germany. [4] Department of Rheumatology, Sahlgrenska University Hospital, Gothenburg, Sweden. [5] Department of Clinical Immunology and Transfusion Medicine, Sahlgrenska University Hospital, Gothenburg, Sweden. [6] Institute of Medical Microbiology, University Hospital of Münster, Münster, Germany. [7] These authors contributed equally: Pradeep Kumar Kopparapu, Patrick Ebner. ✉email: tao.jin@rheuma.gu.se

Septic arthritis is the most dangerous joint disease causing bone destructions in the course of days[1]. The prognosis of septic arthritis is poor, as up to 50% of patients with optimal antibiotic treatments have permanent joint dysfunction[2]. Staphylococcus aureus (S. aureus) is the most common cause of septic arthritis. Innate immunity, such as neutrophils[3] and complement system[4], is crucial for controlling initiation and development of septic arthritis.

S. aureus manipulates the immune system by producing large numbers of virulence factors including toxins, thus leading to hyper-responsiveness or immune evasion[5]. Peptides defined as phenol-soluble modulins (PSMs) constitute one of the most potent group of toxins that are generated by staphylococci[6]. Among these, especially peptides belonging to the PSMα sub-group are known to exert potent biological functions and play roles in the pathogenesis of staphylococcal infections[7]. At nanomolar concentrations, PSMα peptides modulate dendritic cells (DCs) activation and promote neutrophils activation via the formyl peptide receptor (FPR) 2[8–10]. This activation of the chemoattractant FPR2 is achieved without any induction of chemotaxis[11]. At higher concentrations, PSMα peptides are membrane active and mediate lysis of red blood cells, osteoblasts and leukocytes[12–15], especially in their apoptotic state[9]. For PSMα3 it has been shown that it penetrates and modulates human monocyte-derived DCs by altering the TLR2- or TLR4-induced maturation, inhibits pro- and anti-inflammatory cytokine production, induces tolerogenic DCs from healthy donors, and even enhanced differentiation of CD4+ T cells from patients with Th17-associated autoimmune diseases to $T_{regs}$[16]. Just like many other staphylococcal exotoxins, PSMs are positively regulated by the accessory gene regulator (agr), thus connecting the response to quorum sensing with virulence[13,17]. Inactivation of PSMα strongly reduces S. aureus virulence in mice[13], and the immunomodulatory functions of PSMα peptides have been shown to be sensitive to the reactive oxygen species (ROS) that are generated by the neutrophil myeloperoxidase (MPO)-$H_2O_2$[9]. In addition, PSMα peptides simultaneously drive neutrophils into a suppressed state that is characterized by a diminished response to surface receptor independent stimuli such as the phorbol ester, Phorbol myristate acetate (PMA)[9].

In a mouse model of osteomyelitis, PSMα peptides have been shown to be responsible for remodeling and destruction of the bone, effects that are mediated by the cytotoxicity of PSMα on osteoblasts[14]. Based on the fact that biofilm-like structures are often observed in joint infections, bacterial bone/joint infections are considered to be linked to biofilm formation[18]. PSMs are known to structure the biofilms that are formed by S. aureus bacteria and to cause biofilm detachment and bacterial dissemination in a quorum-sensing controlled fashion[19]. It is therefore possible that PSMs have impact on the disease outcome of joint infections by controlling biofilm infection. So far, it is still largely unknown whether expression of PSMs contribute to the development of hematogenous S. aureus septic arthritis.

Historically, there has been a tremendous research focus on the role of PSMα in disease and hence, very little is known about the role/function of PSMβ. This is probably due to the fact that compared to PSMα peptides, PSMβ is a fairly weak neutrophil activating ligand[10]. Also, a low level of PSMβ expression in S. aureus has been shown not to affect the disease activity in a sepsis model[13]. To our surprise, in the current study we found that deletion of PSMβ in the S. aureus Newman strain gave rise to hypervirulent bacteria that caused more severe septic arthritis and worsened weight development in infected animals, compared to animals infected with the parental strain. In contrast, deletion of PSMβ had no impact on the mortality in a sepsis model. We also show that PSMα and PSMβ peptides possess distinct biological functions in vitro, with PSMβ blocking the oxygen radical release that is induced by PSMα in neutrophils and partially reversing the inhibitory effect of PSMα on the PMA induced neutrophil response.

## Results

**PSMβ1 is a conditional neutrophil agonist.** To compare the capacity of PSMα and PSMβ to activate neutrophils, we measured the neutrophil NADPH-oxidase activity following exposure to different concentrations of the two synthetic PSM peptides (Fig. 1a, b). The neutrophil response to PSMα3 was rapidly initiated, reached a maximum value after around 1 min and was of the same magnitude as that induced by the prototype peptide agonist WKYMVM, a specific ligand of FPR2 (Fig. 1a). In contrast, no neutrophil response was induced by PSMβ1, not even with 10-times higher concentration (500 nM; Fig. 1a, b). In order to obtain a response using PSMβ1, the neutrophils had first to be sensitized/primed with tumor necrosis factor-α (TNF-α) (Fig. 1c).

In agreement with earlier findings[11], the PSMα3-induced response was fully inhibited by the FPR2 specific antagonist PBP10 and was unaffected by the FPR1 specific antagonist cyclosporin H (CysH) (Fig. 1d). Interestingly, the NAPDH-oxidase activity induced by PSMβ1 was partially inhibited by PBP10 and CysH (Fig. 1d), suggesting that PSMβ1 is a dual FPR1/FPR2 agonist. As expected, positive controls were fully inhibited by respective antagonists, as fMLF (FPR1 agonist) was blocked by CysH and WKYMVM was totally inhibited by PBP10.

**PSMβ1 inhibits PSMα3-induced release of oxygen radicals from neutrophils.** As both PSMβ1 and PSMα3 share the same receptor - FPR2 - to induce superoxide release from neutrophils, and PSMβ1 has poor activity to neutrophils, we hypothesized that PSMβ1 might counteract the ability of PSMα3 to induce oxygen radical release by neutrophils. To test this, we pre-incubated the neutrophils with PSMβ1 and added PSMα3 (50 nM) as the second stimulus. Strikingly, PSMβ1 fully inhibited PSMα3 induced superoxide release by neutrophils (Fig. 2a, b). To investigate whether PSMβ1 inhibits PSMα3 in a dose-dependent manner, neutrophils were pre-incubated with PSMβ1 in different concentrations (0-500 nM) followed by PSMα3 stimulation (Fig. 2c). Indeed, PSMβ1 dose-dependently inhibited PSMα3-induced superoxide release by neutrophils and the inhibitory effect was observed at concentrations equal or higher than 125 nM (Fig. 2c).

**PSMβ1 inhibits F2Pal10-induced but not WKYMVM-induced release of oxygen radicals from neutrophils.** To further understand how the inhibitory effect of PSMβ1 on PSMα3 is mediated, we pre-incubated neutrophils with PSMβ1 and stimulated the cells with WKYMVM and F2Pal10, which are well-established, FPR2 specific peptide agonists. PSMβ1 did not exert any inhibitory effect on WKYMVM (Supplementary Fig. 1a). In contrast, neutrophils pre-incubated with PSMβ1 released significantly lower oxygen radical when they were exposed to high concentrations (500 nM) of F2Pal10 compared to the cells that had not been pre-incubated with PSMβ1 (Supplementary Fig. 1b). This discrepancy might be due to the fact that these agonists have different binding sites on FPR2. F2Pal10 but not WKYMVM may share the similar binding site as PSMβ1.

**PSMβ1 protects neutrophils from the inhibitory effect of PSMα3 on the neutrophil response to PMA.** PMA, a potent activator of the protein kinase C, activates the neutrophil NADPH oxidase without the involvement of any surface receptor[20]. The neutrophil response to PMA is largely reduced by high concentrations of PSMα peptides (shown for PSMα3 in

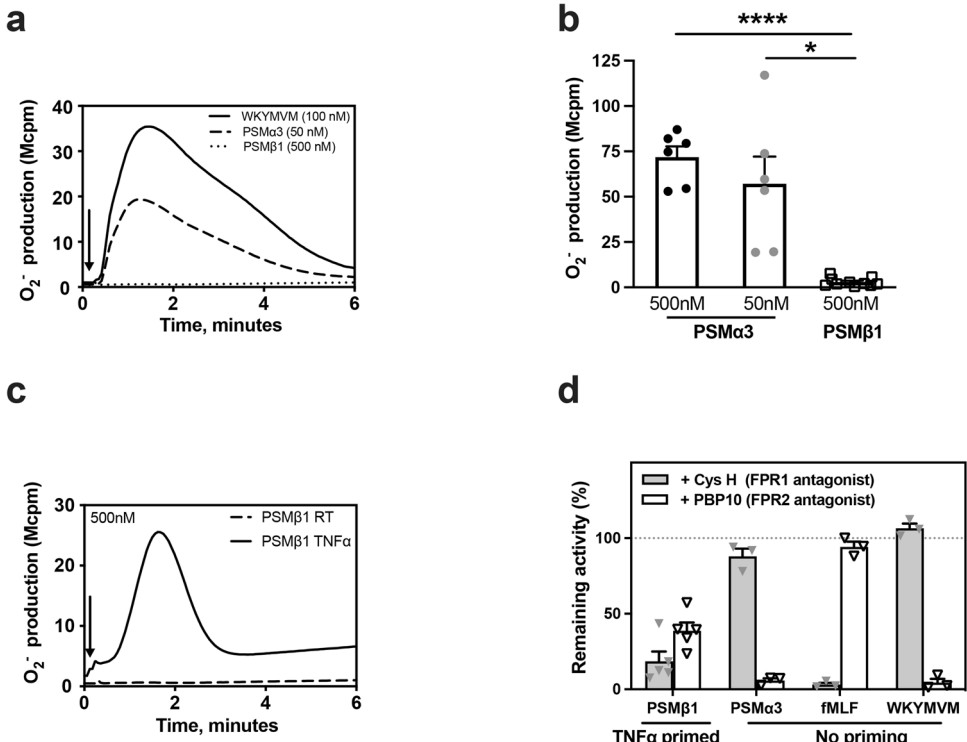

**Fig. 1 PSMβ is a conditional neutrophil agonist. a** NADPH oxidase–mediated superoxide anion release by neutrophils was measured by isoluminol-amplified chemiluminescence. The cells were preincubated at 37 °C for 5 min before being challenged with agonist (arrow to the left) and measurement of superoxide anion release ($O_2^-$, y-axis) over time (min), one representative trace out of six to ten individual experiments for neutrophils stimulated with PSMα3 (50 nM), PSMβ1 (500 nM) or WKYMVM (control, 100 nM) is shown. **b** The peak $O_2^-$ release by neutrophils stimulated with the following agonists: PSMα3 (50 nM, $n = 6$), PSMα3 (500 nM, $n = 6$) or PSMβ1 (500 nM, $n = 10$) was compared. **c** The cells were primed with TNF-α (10 ng/ml, 37 °C, 20 min) or not (PSMβ1 RT, dashed line), then preincubated at 37 °C for 5 min before being challenged with PSMβ1 (500 nM) (arrow to the left) and measurement of superoxide anion release ($O_2^-$, y-axis) over time (min). One representative trace out of five individual experiments is shown. **d** The cells were primed with TNF-α (10 ng/ml, 37 °C, 20 min) or not as indicated, then incubated in the absence (as control) or presence of the FPR2 antagonist PBP10 (1 μM) or the FPR1 antagonist Cyclosporin H (1 μM, CysH) for 5 min at 37 °C before challenge with an agonist and measurement of superoxide anion release ($O_2^-$, y-axis) over time (min). The peak $O_2^-$ release by neutrophils stimulated with the following agonists: PSMβ1 (500 nM, $n = 5$), PSMα3 (50 nM, $n = 3$), fMLF (100 nM, $n = 3$), or WKYMVM (100 nM, $n = 3$), in the absence (100% control) or presence of the FPR1 inhibitor CysH (black bars) or the FPR2 inhibitor PBP10 (gray bars) was compared. The bar graph shows the percent remaining $O_2^-$ activity for each agonist in the presence of the antagonists. Statistical comparison was done using paired t test, with data expressed as mean ± standard error of the mean **b**, **d**. *$P < 0.05$, ****$P < 0.0001$.

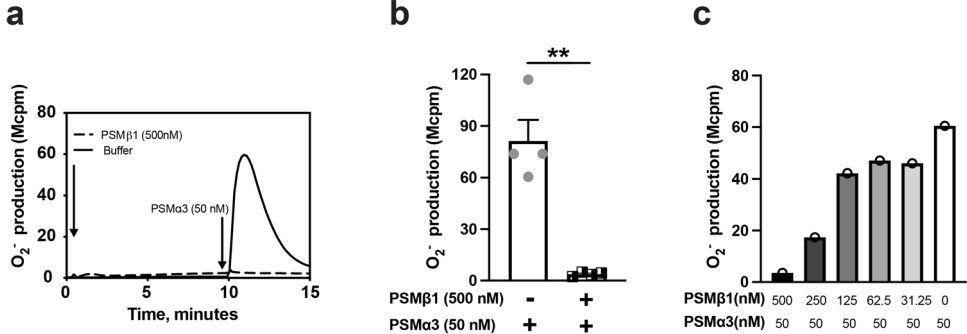

**Fig. 2 The PSMα3 induced activation of the neutrophil NADPH-oxidase is inhibited by PSMβ1. a** NADPH oxidase-induced superoxide anion release ($O_2^-$, y-axis) by neutrophils was measured by isoluminol-amplified chemiluminescence over time (min, x-axis). The cells were preincubated at 37 °C for 5 min before being first challenged with one agonist (arrow to the left) and then with a second agonist PSMα3 (50 nM, arrow to the right) once the first response had returned to baseline. Shown is one representative trace out of four individual experiments for neutrophils first stimulated with PSMβ1 (500 nM) or buffer (control) and then PSMα3 (50 nM). **b** The peaks $O_2^-$ release by neutrophils stimulated with PSMβ1 (500 nM) or buffer control followed by PSMα3 (50 nM) were compared. **c** The representative bar graphs show the peak $O_2^-$ release from the neutrophils first stimulated with different concentrations of PSMβ1 (500 nM, 250 nM, 125 nM, 62.5 nM, 31.25 nM, 0 nM), and then challenged with PSMα3 (50 nM). The experiment was performed 3 times with different buffy coats. All experiments showed the similar pattern. Statistical comparison was done using paired t test, with data expressed as mean ± standard error of the mean **b**. **$P < 0.01$.

**a**

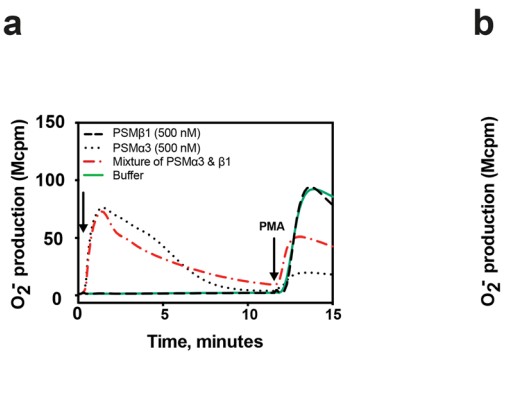

**b**

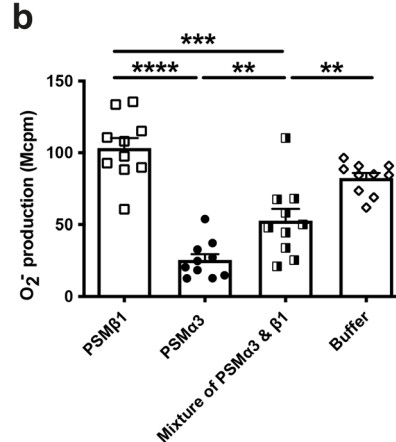

**Fig. 3 PSMβ1 protects neutrophils from the inhibitory effect of PSMα3 on the neutrophil response to PMA. a** NADPH oxidase–mediated superoxide anion release by neutrophils was measured by isoluminol-amplified chemiluminescence. The cells were preincubated at 37 °C for 5 min before being challenged with the first agonist (arrow to the left) followed by a second stimulus (PMA, 50 nM) once the first response had returned to baseline. Shown is one representative trace out of 10 individual experiments for neutrophils stimulated with PSMβ1 (500 nM), PSMα3 (500 nM), mixture of PSMα3 and PSMβ1 (500 + 500 nM) or buffer control and followed by PMA stimulation. **b** The peak $O_2^-$ release by neutrophils stimulated with PMA following stimulation of PSMβ1 (500 nM), PSMα3 (500 nM), mixture of PSMα3 and PSMβ1 (500 + 500 nM) or buffer control was compared in 10 experiments. Statistical comparison was done using paired t test, with data expressed as mean ± standard error of the mean **b**. **$P < 0.01$; ***$P < 0.001$; ****$P < 0.0001$.

Fig. 3a), and this reduction is FPR2-independent[11]. As PSMβ1 inhibits the receptor dependent activity of PSMα3, we further studied whether PSMβ1 was also able to antagonize the receptor independent effect of PSMα3. PSMβ1 alone had no effect on the neutrophil response to PMA, but when combined with PSMα3, it reduced the inhibitory effect of the latter (Fig. 3a, b) showing that PSMβ1 protects neutrophils from the inhibition mediated by PSMα3.

**PSMβ deficient _S. aureus_ caused a more severe septic arthritis**. To study the role of PSMs in septic arthritis, PSMα or β deficient _S. aureus_ Newman strains were used in a well-established mouse model of septic arthritis (Fig. 4). Mice injected with the Δ_psmβ_ mutant developed significantly more severe clinical arthritis than mice infected with the wild type (WT) or Δ_psmα_ strain. The difference between the groups was observed already on day 2 after bacterial inoculation and continued until the end of experiment on day 10 (Fig. 4a).

Not only the severity but also the frequency of arthritis was significantly higher in mice infected with Δ_psmβ_ strain (Fig. 4b). On day 3, 88% of the mice in the Δ_psmβ_ group developed arthritis, whereas the arthritis frequency in the Δ_psmα_ and Newman group was 46% and 43%, respectively. At the end of the experiment, on day 10 post-infection, the Δ_psmβ_ injected group had an arthritis frequency of 93% as compared to 69% in the Δ_psmα_ group and 63% in the Newman group (Fig. 4b). Moreover, there was a clear trend towards an increased frequency of clinical polyarthritis (arthritis in ≥2 joints) in Δ_psmβ_ infected mice compared with others (Fig. 4c). As expected, the Δ_psmα_ infected mice had the least weight loss percentage among the groups. In contrast, Δ_psmβ_ infected mice had a significantly worse weight development compared with WT infected mice (Fig. 4d).

**PSMβ deficiency results in increased bone erosion in septic arthritis**. To further validate the clinical arthritis data, we performed micro-computed tomography (μCT) scans of joints to analyze the bone erosion. There was significantly more damage of the bone in the Δ_psmβ_ infected mice compared with the WT and Δ_psmα_ infected mice (Fig. 5a), whereas there was no significant difference between mice infected with parental and Δ_psmα_ mutant (Fig. 5a). Similarly, the highest frequency of bone damage in

was found in mice infected with Δ_psmβ_ (Fig. 5b). Figure 5c–g show the representative images of bone damage in the wrist (Fig. 5c), elbow (Fig. 5d), hip (Fig. 5e), knee (Fig. 5f) and feet (Fig. 5g), respectively.

**Bacterial clearance in kidneys and joints is attenuated by PSMα**. Δ_psmα_ infected mice had significantly lower kidney abscess scores than WT or Δ_psmβ_ infected mice on both day 3 and day 10 post-infection (Fig. 6a). Moreover, Δ_psmα_ infected mice presented significantly lower bacterial counts in their respective kidneys as compared to mice infected with Δ_psmβ_ and its parental strain on both day 3 and day 10 post-infection (Fig. 6b), suggesting that PSMα attenuates the bacterial clearance capacity of the host. Interestingly, higher frequency of bacterial persistence in the joints were significantly higher in both WT and Δ_psmβ_ infected mice than Δ_psmα_ infected mice on day 3 post-infection (Fig. 6c), suggesting that the WT and Δ_psmβ_ strains has a better ability to reach the joint cavity at the early phase of disease (day 3) than the Δ_psmα_ strain.

**PSMβ deficiency upregulates serum cytokines in mice at the early phase**. We further studied the neutrophil-related cytokine/chemokine levels during the course of infection. Significantly lower interleukin 6 (IL-6) and keratinocyte-derived chemokine (KC) levels were found in the Δ_psmα_ infected mice compared with both WT and Δ_psmβ_ infected mice on day 3 after infection. Interestingly, both IL-6 and KC levels were significantly higher in Δ_psmβ_ infected mice than WT strain infected mice, suggesting Δ_psmβ_ infected mice had more severe systemic infection than WT infected mice. However, at the late time point (day 10 post-infection) no significant difference was observed in IL-6 and KC levels (Fig. 7a, b).

**Neutrophil apoptosis and death in _S. aureus_ PSM mutants infected mice**. We further assessed the impact of PSMs on apoptosis and death of neutrophil in mice infected with PSM mutants at the early time point (day 3 post-infection). Figure 8a demonstrates the gating strategy for neutrophils and the representative FACS plots for Annexin V and 7-aminoactinomycin D (7-AAD) staining of blood neutrophils from mice infected with _psm_ mutant strains. As expected, the percentage of neutrophils in

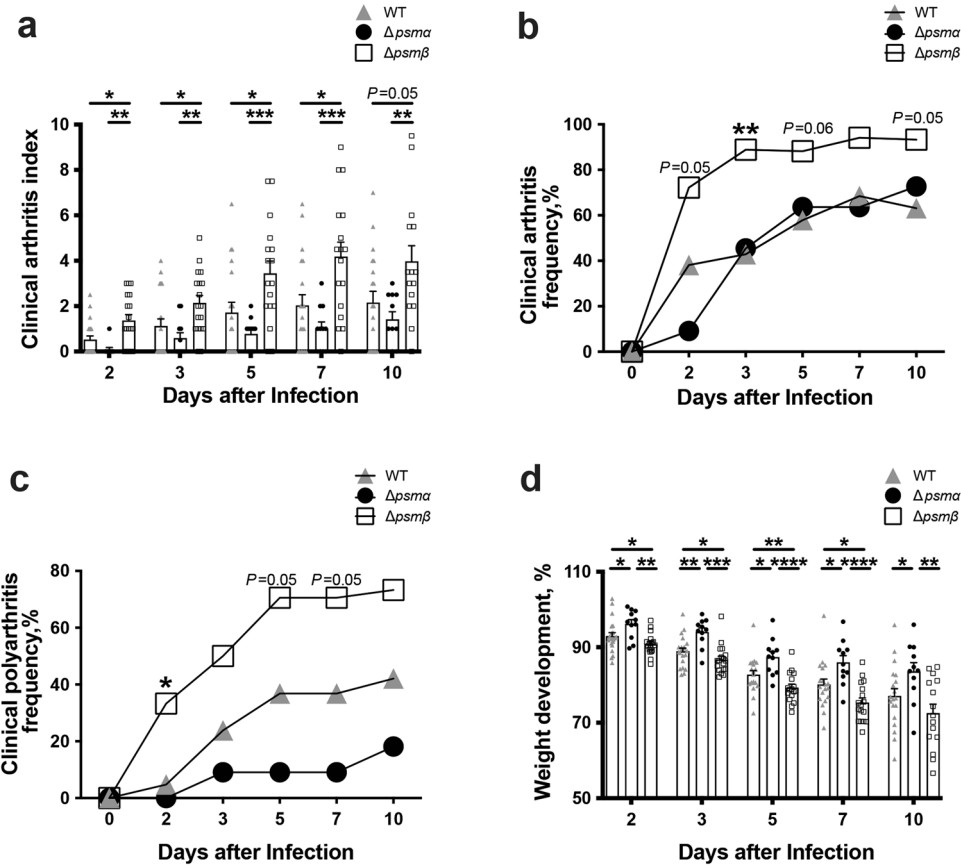

**Fig. 4 PSMβ deficient mutant is hypervirulent in a murine model of septic arthritis.** NMRI mice ($n = 11$–21/group) were intravenously injected with *S. aureus* Newman (wild type, WT), or the isogenic mutant strains Δ*psmα* or Δ*psmβ* ($5 \times 10^6$ CFU/mouse) and sacrificed on day 10 post-infection. The arthritis severity **a**, frequency of arthritis **b**, frequency of polyarthritis **c**, and the changes in the body weight **d** were monitored for 10 days post-infection. The results from three independent experiments were pooled. Statistical comparison was performed using the Mann-Whitney *U* test **a**, **d**, and Fisher's exact test **b**, **c**. Data are expressed as mean ± standard error of the mean **a**, **d** or percentage **b**, **c**. *$P < 0.05$; **$P < 0.01$; ***$P < 0.001$; ****$P < 0.0001$.

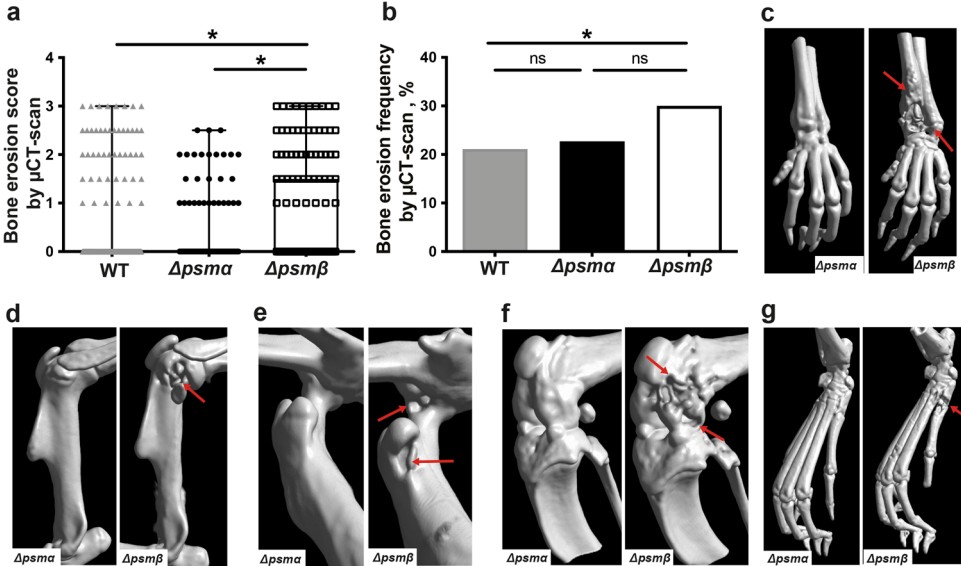

**Fig. 5 PSMβ deficient mutant strain causes more severe joint destruction.** NMRI mice ($n = 11$–21/group) were intravenously injected with *S. aureus* Newman (wild type, WT), or the isogenic mutant strains Δ*psmα* and Δ*psmβ* ($5 \times 10^6$ CFU/mouse) and sacrificed on day 10 post-infection. Shown is the accumulative bone destruction scores **a** and frequency of bone destructions **b** of the joints from all 4 limbs of mice as assessed by micro computed tomography (μCT) scan. Representative μCT scan images **c**–**g** showing both intact (left) and heavily eroded (right) joints, **c** wrists, **d** shoulders, **e** hips, **f** knees and **g** hind paws. Arrows indicate the bone erosion. The results from three independent experiments were pooled. The data are reported as mean ± standard error of the mean and analyzed with the Mann-Whitney *U* test, with data expressed as box plots showing interquartile range, and whiskers showing minimum and maximum **a** or Fisher's exact test **b**. *$P < 0.05$; ns not significant.

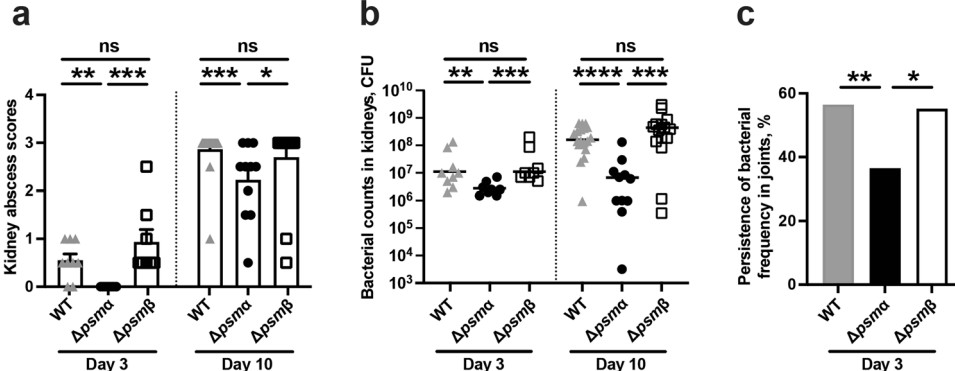

**Fig. 6 Bacterial clearance in kidneys and joints is attenuated by PSMα.** NMRI mice (*n* = 8–21/group) were intravenously injected with *S. aureus* Newman (wild type, WT), or the isogenic mutant strains Δ*psmα* or Δ*psmβ* (5 × 10⁶ CFU/mouse) and sacrificed on day 3 or day 10 post-infection. **a** Kidney abscess scores and **b** persistence of *S. aureus* in kidneys 3 days and 10 days after infection. The results from four independent experiments were pooled. **c** Persistence of *S. aureus* frequency in joints including shoulders, elbows, front paws, hips, knees, and hind paws of the mice 3 days after infection. Statistical evaluations were performed using the Mann-Whitney *U* test **a**, **b**, or Fisher's exact test **c**. Data are presented as the mean ± standard error of the mean **a** or median **b**. *$P < 0.05$; **$P < 0.01$; ***$P < 0.001$; ****$P < 0.0001$; ns not significant.

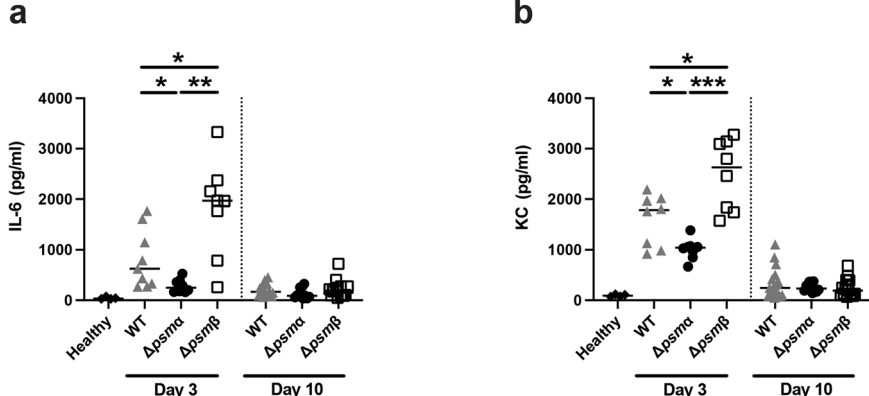

**Fig. 7 PSMβ deficiency upregulates serum cytokines in mice at the early phase.** NMRI mice (*n* = 8–21/group) were intravenously injected with *S. aureus* Newman (wild type, WT), or the isogenic mutant strains Δ*psmα* or Δ*psmβ* mutants (5 × 10⁶ CFU/mouse) and sacrificed on day 3 or day 10 post-infection. Serum levels of **a** interleukin 6 (IL-6) and **b** keratinocyte-derived chemokine (KC) were analyzed. The results from four independent experiments were pooled. The data are reported as mean ± standard error of the mean and analyzed with the Mann-Whitney *U* test. *$P < 0.05$; **$P < 0.01$; ***$P < 0.001$.

the leukocytes was increased in all infected mice compared to healthy controls (Fig. 8b). Significantly higher percentage of neutrophils was found in Δ*psmα* infected mice compared to Δ*psmβ* or WT infected mice. In line with these data, further analyses of neutrophil apoptosis and death revealed that both Δ*psmβ* and WT infected mice had more dead neutrophils (Annexin V - and 7-AAD +) than Δ*psmα* infected mice. Interestingly, Δ*psmβ* infected mice tended to have more dead neutrophils than WT infected mice in blood (Fig. 8c), suggesting that PSMβ may protect the neutrophils from cytotoxic effects of PSMα. In contrast, the percentage of apoptotic neutrophils at the end stage (Annexin V + and 7-AAD +) were higher in Δ*psmα* infected mice than Δ*psmβ* or WT infected mice (Fig. 8d). No difference was observed in the percentage of apoptotic neutrophils at the early stage (Annexin V + and 7-AAD - ) among the groups (Fig. 8e).

**Δ*psmβ* strain caused similar mortality as its parental strain**. To further understand whether deletion of PSMβ had any impact on mortality rate in *S. aureus* sepsis, the mice were infected with Δ*psmβ* or its parental strain in sepsis doses. One hundred percent mortality was observed in both groups by day 5 (Supplementary Fig. 2), suggesting that PSMβ depletion did not have impact on the outcome of sepsis.

**Expression of virulence factors in *S. aureus* PSM mutants and their parental strain**. Surface proteins including protein A, clumping factors and von Willebrand binding proteins are known to be crucial virulence factors in *S. aureus*-induced septic arthritis. To rule out the possibility that deletion of PSMs influences expression of those surface proteins, relative expression of surface proteins in the PSM mutants was compared with that of their parental strain after 2, 6, and 24 h of in vitro bacterial growth. There were no statistically significant differences among the groups (Supplementary Fig. 3) indicating that deletion of PSMs does not affect the surface expression of the studied virulence factors. To understand whether deletion of *psm* genes influences the expression of alpha-hemolysin that is a major virulence factor, we used the traditional blood agar plate hemolysin assay and found that the hemolysis was similar in those three strains (Supplementary Fig. 4).

**Expression of PSMα in *S. aureus* PSM mutants and its parental strain**. To exclude the possibility of PSMα upregulation in the Δ*psmβ* strain, the gene and protein expression levels of these two mutant strains and WT were analyzed by qRT-PCR and HPLC (Supplementary Fig. 5). As expected, no PSMα expression was detected in the Δ*psmα* strain. Relative gene expression of PSMα was significantly increased in the Δ*psmβ* strain compared with the

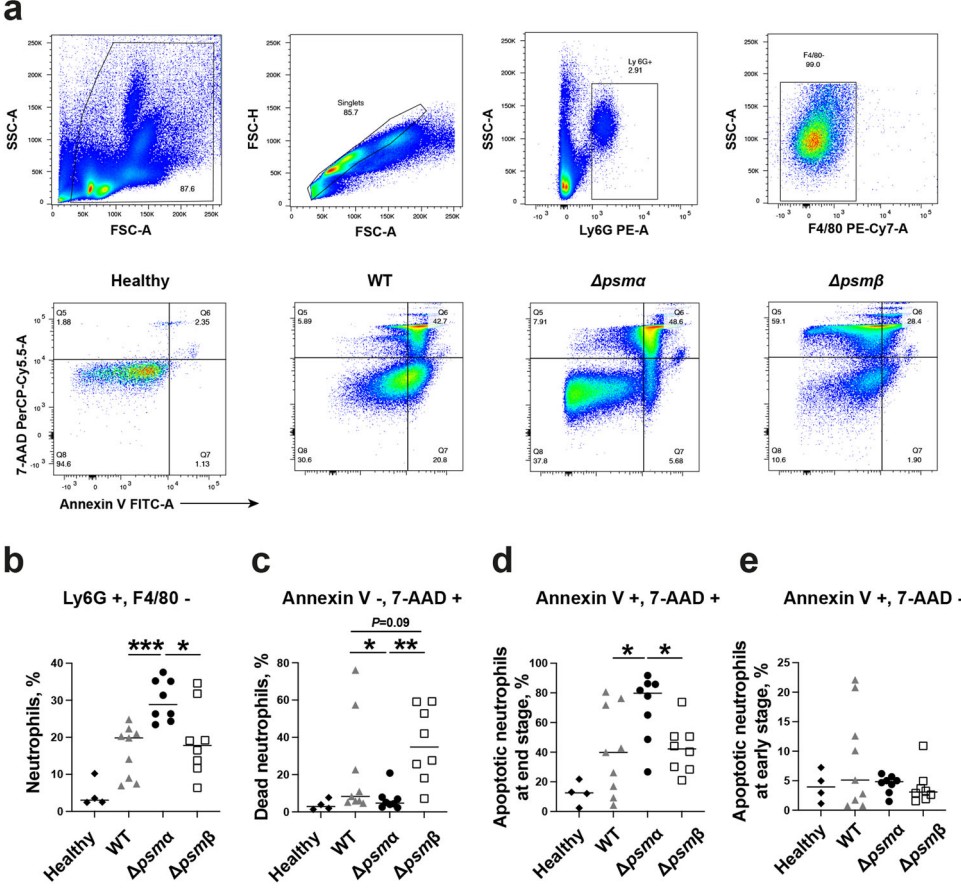

**Fig. 8 Neutrophil apoptosis and death in *S. aureus* PSM mutants infected mice.** Blood samples were collected for flowcytometry analyses from NMRI mice ($n = 8$–9/group) after intravenous infection with *S. aureus* Newman (wild type, WT), or the isogenic mutant strains $\Delta psm\alpha$ or $\Delta psm\beta$ mutants ($4.1$–$5.4 \times 10^6$ CFU/mouse) on day 3 post-infection. **a** Representative images of gating strategy for neutrophils (upper panel) and apoptosis/death of neutrophils (lower panel). Annexin V indicate apoptotic cell death, while 7-aminoactinomycin (7-AAD) stained nuclei is considered as necrotic cell death. **b** Neutrophils are gated as Ly6G⁺/F4/80⁻. **c** Dead neutrophils are gated as Annexin V⁻/7-AAD⁺. **d** Apoptotic neutrophils at end stage are gated as Annexin V⁺/7-AAD⁺. **e** Apoptotic neutrophils at early stage are based on Annexin V⁺/7-AAD⁻. The results from two independent experiments were pooled. Data are presented as the median and analyzed with the Mann-Whitney $U$ test. *$P < 0.05$; **$P < 0.01$; ***$P < 0.001$.

WT strain only at early (2 h) but not later time points (6 and 24 h) of bacterial culture (Supplementary Fig. 5a). Moreover, the PSMα peptide was also identified in the supernatant of WT, $\Delta psm\alpha$, $\Delta psm\beta$ strains at 3 different time points by HPLC (Supplementary Fig. 5b, c). These data clearly indicate that while the $\Delta psm\alpha$ strain cannot produce PSMα, there is no altered level of PSMα in the WT and $\Delta psm\beta$ strain in vitro.

## Discussion

The biological function of PSMα has been extensively studied. The release of PSMα by *S. aureus* causes cell lysis of leukocytes, resulting in immune evasion and bacterial overgrowth in the infected organs. In contrast, PSMβ, double size of PSMα from the very same family, has been much less investigated. In this study, we reveal that PSMβ is a poor neutrophil activator compared to PSMα. PSMβ blocks the PSMα induced release of oxygen radicals from neutrophils and partially inhibits the cytotoxic effect of PSMα. Importantly, a PSMβ deficient *S. aureus* strain is hypervirulent in a mouse model of septic arthritis, suggesting that PSMβ might be protective in *S. aureus* septic arthritis.

PSMα is multifunctional depending on the concentrations. At lower concentrations PSMα activates neutrophils to produce superoxide through FPR2, which might be beneficial for the host to eliminate the invading *S. aureus*[21]. At higher concentrations PSMα exerts a cytolytic effect on neutrophils, especially apoptotic

cells, independently of FPR2[9], which impairs the first-line immune defense. At the same time, PSMα but not PSMβ is known to boost excretion of *S. aureus* cytoplasmic proteins[22] that might directly contribute to the pathogenicity of *S. aureus*[23]. It seems that the detrimental effect of PSMα in the course of *S. aureus* infections is predominant compared to the protective effect prior to the onset of infection, as PSMα deficient strain caused less severe systemic infection and less bacterial load in the kidneys in both current study and previous report[13].

The 3-D structures of PSMα and PSMβ are very different[24], which may explain their distinct biological functions. PSMβ was shown to possess the chemoattractant capacity and activate neutrophils at relatively high concentration[10]. In the current study, we demonstrated that PSMβ is a very poor neutrophil stimulator compared to PSMα, as only TNF-primed neutrophils responded significantly to the stimulation of PSMβ. Interestingly, oxygen radical release of neutrophils upon PSMα stimulation was abolished by pre-incubation with PSMβ in a dose-dependent manner. We hypothesize that the inhibitory effect of PSMβ on PSMα activity is mediated through the FPR2, as our data suggest that both PSMα and PSMβ share the same receptor- FPR2 – to activate neutrophils, although PSMβ can also mediate its effects through FPR1. It is very likely that PSMβ competitively binds to FPR2 without activating neutrophils and thus blocks the neutrophil stimulating effect of PSMα.

Hypervirulence of the Δpsmβ strain has been reported before in a mouse model of *S. aureus* skin infection[13]. In the current study, we show that deletion of *psmβ* led to development of more severe septic arthritis, whereas the bacterial load in organs and mortality in a sepsis model of infection was similar in the Δpsmβ and the parental strain. Why is the *psmβ* depleted strain more virulent in inducing septic arthritis caused by hematogenous spreading? *S. aureus* needs to survive in the blood stream and invade the joint cavity to induce the septic arthritis. Neutrophils are one of the main house keepers to eliminate the invading bacteria and frequency of neutrophils increase in the blood after infection. Interestingly, on day 3 post-infection Δpsmβ and the parental strain infected mice had significantly less neutrophils and more dead neutrophils than Δpsmα infected mice, strongly suggesting that expression of PSMα by *S. aureus* causes the reduced neutrophil by inducing cell death. Notably, Δpsmβ infected mice tended to have more dead neutrophils than parental strain infected mice, hinting that PSMβ may protect the cell death of neutrophils caused by PSMα. There are several possible mechanisms of neutrophil death caused by PSMα. Firstly, at the high concentration PSMα is cytotoxic to neutrophils; Secondly, PSMα induces neutrophil necroptosis and contribute to MRSA pneumonia[25]; Thirdly, PSMα selectively permeabilizes apoptotic neutrophils that are usually increased in infections[9,26]; Finally, it is known that neutrophils initiate mechanisms of cell death as soon as they are activated and start to perform their effector functions[27]. We speculate that the inhibitory effect of PSMβ on neutrophil activation by PSMα reduce the cell death and maintain sufficient number of functional neutrophils. This may explain why Δpsmβ strain is hypervirulent in inducing septic arthritis, as neutrophil depletion was shown to greatly aggravate the septic arthritis[3].

Surface proteins such as clumping factors and von Willebrand binding protein are known to facilitate the bacteria-host interaction and consequently promote the joint invasiveness of *S. aureus*[28,29]. Our data demonstrated that the expression of several surface proteins responsible for septic arthritis development was similar in PSMs mutants and their parental Newman strain. This ruled out the possibility that PSMβ depletion promotes the gene expression of surface proteins. However, we cannot fully exclude that possibility that PSMβ depletion had the post-transcriptional impact on the surface proteins and thereby underlies the increased virulence of the *psmβ* mutant strain. We also hypothesized that Δpsmβ may overexpress PSMα, thus leading to hypervirulence of the strain. The relative gene expression levels of PSMα in the Δpsmβ strain was indeed significantly higher than its parental strain at early time point (2 h) but not at later time points. Also, PSMα levels in the culture filtrate were also higher in Δpsmβ strain than WT strain, suggesting compensatory overexpression of PSMα in Δpsmβ strain. However, the clinical significance of subtle overexpression of PSMα at early time point in the PSMβ deficient strain is somehow uncertain. Since the deletion of PSMα in *S. aureus* had no impact on the development of septic arthritis, the hypervirulence of PSMβ deficient strain in septic arthritis may not be explained by compensatory overexpression of PSMα.

Expression of PSMs in *S. aureus* has a subtle connection with *S. aureus* lipoproteins (Lpps), which are bacteria-derived ligands of the toll-like receptor 2 (TLR2) and largely involved in the pathogenesis of various *S. aureus* infections[30–33]. The membrane damaging activity of PSMα promotes the shedding of Lpps from the bacterial cytoplasmic membrane, thus triggering the activation of innate immunity via TLR2[34]. PSMα but not PSMβ is involved in this process by controlling the release and disruption of *S. aureus* extracellular vesicles[35,36]. One can speculate that PSMβ expression in *S. aureus* may also antagonize the PSMα-dependent production of extracellular vesicles and the release of Lpps, consequently attenuating TLR2 mediated inflammation during *S. aureus* infections. Indeed, the levels of IL-6 (a proinflammatory cytokine) and KC (a neutrophil chemoattractant) were significantly higher in the Δpsmβ strain infected mice than WT strain infected mice at the early phase of disease. This indicates that Δpsmβ strain caused more severe disease and systemic inflammation than WT and Δpsmα strains. These findings are also in agreement with our previous data that serum levels of IL-6 were significantly correlated with the severity of joint destruction in septic arthritis[1].

It still remains elusive why *S. aureus* produce both PSMs and what is the advantage for them to produce two antagonist molecules during an infection. As PSMα is capable to non-specifically damage cytoplasmic membrane of *S. aureus*[22], we speculate that PSMβ expression may also protect *S. aureus* from cell damage of PSMα, whereas PSMα is produced to colonize, spread to other tissues and escape from the immune system. More studies are warranted to answer above questions.

In summary, our data demonstrate distinct roles of PSMα and β expression in *S. aureus* septic arthritis. PSMα impaired host immune killing but had no impact on the induction of septic arthritis, whereas PSMβ expression protected from the development of septic arthritis. In vitro, PSMβ blocked the PSMα induced release of oxygen radicals by neutrophils and partially inhibited the cytotoxic effect of PSMα.

## Materials and methods

**Chemicals**. PSMα3 and PSMβ1 were purchased from EMC (Tübingen, Germany), Isoluminol and the FPR agonist formyl-Met-Leu-Phe (fMLF) were purchased from Sigma-Aldrich (St. Louis, MO, USA), Cyclosporin H (CysH; antagonist for FPR1) was kindly provided by Novartis Pharma (Basel, Switzerland). The hexapeptide WKYMVM (agonist for FPR2) was purchased from Alta Bioscience. The FPR2-specific antagonist PBP10 (gelsolin residues 160–169[37]) as well as the F2Pal10 pepducin were purchased from CASLO Laboratory (Lyngby, Denmark). HRP was purchased from Roche Diagnostics (Bromma, Sweden). TNF-α, phorbol 12-myristate 13-acetate (PMA) were purchased from MilliporeSigma (Burlington, MA, USA).

All peptides were dissolved in DMSO, and subsequent dilutions of peptides and other reagents were made in Krebs–Ringer phosphate buffer (KRG; 10 mM glucose, 1.5 mM $Mg^{2+}$, and 1 mM $Ca^{2+}$ in $dH_2O$, pH 7.3).

**Isolation of human neutrophils**. Buffy-coats from healthy donors were obtained from the blood bank at Sahlgrenska University Hospital. Since the buffy coats were provided anonymously and could not be traced back to a specific individual, ethics approval was not needed. Neutrophils were isolated from these buffy coats using dextran sedimentation and Ficoll-Paque gradient centrifugation, as described by Bøyum[38]. Remaining erythrocytes were removed by hypotonic lysis and the neutrophils were washed and resuspended in KRG ($1 \times 10^7$/ml) and stored on ice until use. In some of the experiments, neutrophils were incubated with TNF-α (10 ng/ml, 37 °C, 20 min) before utilized. The purity of the neutrophil preparations was routinely >90%.

**Measurements of NADPH oxidase activity**. Isoluminol-amplified chemiluminescence (CL) technique performed in a 6-channel Biolumat LB 9505 (Berthold, Wildbad, Germany) was used to measure the production of superoxide anion by the neutrophil NADPH oxidase as described[39]. A 900 μl reaction mixture containing $10^5$ neutrophils, isoluminol ($2 \times 10^{-5}$ M), and horseradish peroxidase (HRP, 4 U) in KRG was prewarmed (5 min at 37 °C) in disposable 4-ml polypropylene tubes, after which activating ligands (100 μl) were added and the light emission was recorded continuously over time. The results are presented as superoxide production [arbitrary units (AU)] given in light units (megacounts/min; Mcpm) over time (min).

**Construction of *S. aureus* PSM deletion mutants and culture condition**. Deletions of the *psmα* and *psmβ* operon were performed as marker-less deletions using allelic replacement as previously described[40]. Briefly, ≈ 1 kb upstream and ≈ 1 kb downstream of the *psm*-operons were amplified using the primers from[22] (up_fwd_alpha CAGATCTGTCGACGATATCTATATGGCTAA AATTCCAGTTAC up_rev_alpha AATCTTAATGAAATAATTTAAGCGAATTG AATACTTAAAATTC down_fwd_alpha CTTAAATTATTTCATTAAGATTAC CTCCTTTGC down_rev_alpha GGCATGCAAGCTTGATATCTGTCATGCTT GATAATTTCG up_fwd_beta CAGATCTGTCGACGATATCTTGAGGTATGC

TTTGCAACC up_rev_beta TTATATTAGAATTCCATTGAAAACACTC
CTTAAAATTTAAATTTG down_fwd beta TCAATGGAATTCTAATATAAT
AACTAATATTCTTTAAAATAAACTGG down_rev beta GGCATGCAAGCTTG
ATATCGCATCTTTTCGTAGTGTCTTTTTTAC) by PCR and ligated into
pBASE6 using Gibson assembly[41]. *S. aureus* Newman were then transformed with
the resulting plasmids. After construction, all the strains were stored in tryptic soy
broth (TSB) containing glycerol at −80 °C.

For infection experiments, the bacteria were prepared as described[42]. Briefly,
premade batches of bacteria were thawed, washed twice with phosphate-buffered
saline (PBS), and diluted to a desired concentration. Viable counts were confirmed
by quantitative plating of the inoculum on horse blood agar plates for each
experiment.

**RNA extraction, cDNA synthesis and gene expression analysis using real-
time quantitative PCR**. The bacteria were cultured in TSB on a shaker at 37 °C
and harvested at different time points of 2, 6 and 24 h. The pellet was suspended
with 1× Trizol (miRNeasy kit, Qiagen, Hilden, Germany). The suspended samples
were lysed using 0.1 mm glass beads (Glass beads-acid washed, Sigma-Aldrich)
with FastPrep® lysis Instrument (Fastprep-24, MP bio, Santa Ana, USA) for
60 seconds at a frequency speed of 6.5. Chloroform was added at a ratio of 1:5
(Trizol: Chloroform) and the aqueous layer was extracted after centrifugation at a
minimum speed of $12,000 \times g$ for 5 min at 4 °C. The aqueous supernatant was
precipitated by using $1\frac{1}{2}$ part of isopropanol before centrifugation at a minimum
of $12,000 \times g$ for 15 min at 4 °C. The RNA pellet was washed using ice-cold 75%
alcohol and suspended with the nuclease-free water before the RNA concentration
measurement. cDNA synthesis was performed using iScript cDNA synthesis kit
according to the manufacturer's protocol (Bio-Rad, Hercules, USA).

The expression levels of Protein A (Spa), Clumping factor A (ClfA), Clumping
factor B (ClfB), von Willebrand factor-binding protein (vWbp), and PSMα were
analyzed with the TaqMan gene expression assays (Applied Biosystems,
Warrington, UK) (1787866 C10 for Protein A, 1787866 C8 for ClfA, 1787866 C9
for ClfB and 1787866 C11 for vWbp and 1789527 C15 which served as an internal
control) using ViiA 7 Fast real-time PCR system (Applied Biosystems, Warrington,
UK). All samples were run in triplicates and the relative expression was calculated
using the ΔΔCt method.

**High-performance liquid chromatography (HPLC) analysis of PSM peptides**.
*S. aureus* Newman and Δpsmα, Δpsmβ variants were inoculated to an OD600 of 0.1
from an overnight culture in TSB and cultivated at 37 °C. Samples were drawn after
2, 4 and 8 h of incubation and centrifuged for 5 min at $15000 \times g$ at 4 °C. The
corresponding supernatants were filtered through sterile syringe filter with 0.2 μm
pore size (Sarstedt, Germany) prior to be concentrated 4× using speedvac vacuum
concentrator. PSM peptides were separated from the concentrated supernatant by
reversed-phase chromatography using an XBridge C8, 5 μm, 4.6 × 150 mm column
(Waters Corporation, Milford, MA, USA) with a fitted pre column. A linear gra-
dient from 0.1% TFA in water to acetonitrile containing 0.08% TFA for 15 min
with additional 5 min of 100% B at a flow rate of 1 ml/min was used and a 50 μl
sample volume was injected. Peaks were detected at 210 nm. The PSM peptides
were eluted between 14 and 18 min.

**Ethics statement**. Mouse studies were reviewed and approved by the Ethics
Committee of Animal Research of Gothenburg. Mouse experiments were con-
ducted in accordance with recommendations listed in the Swedish Board of
Agriculture's regulations and recommendations on animal experiments.

**Animal experiments**. Female NMRI mice, 6–12 weeks of age were purchased from
Envigo (Venray, the Netherlands). Mice were housed under standard environ-
mental conditions of temperature and light and had free access to laboratory chow
and water in the animal facility of the Department of Rheumatology and Inflam-
mation Research, University of Gothenburg. All the experiments were approved by
the Ethics Committee of Animal Research of the University of Gothenburg, and the
animal experimentation guidelines were strictly followed.

**Experimental protocols for septic arthritis mouse models**. Four separate in vivo
experiments were performed for the staphylococcal septic arthritis studies as
previously described[1,43]. In all experiments, 200 μl with different concentrations of
Staphylococcal suspension in PBS were injected intravenously into the tail vein of
mice. The mice were infected with a desired arthritogenic dose ($5 \times 10^6$ colony-
forming units [CFU]/mouse) of Newman WT strain, Δpsmα strain, or Δpsmβ
strain. At day 3, mice were sacrificed and the blood were collected for flowcyto-
metry analyses, plasma or sera were separated to assess cytokine levels, and four
limbs (including shoulders, elbows, front paws, hips, knees, and hind paws) as well
as the kidneys were collected for examination of bacterial persistence. Furthermore,
at day 10, another set of mice were sacrificed and the serum was collected to assess
cytokine levels, paws were collected for radiological examination of bone erosions,
and the kidneys were removed for the assessment of bacterial persistence.

**Clinical evaluation of septic arthritis**. All 4 limbs of each mouse were individually
evaluated for the development of arthritis by three observers (Z.H., T.J., and M.M.)
who were blinded to the treatment groups. The development of clinical arthritis
was monitored at regular intervals until the end of study. Arthritis was defined as
visible joint swelling or erythema of the joints and paws. To evaluate the intensity
of arthritis, a clinical scoring system of 0–3 points for each limb was carried
out[1,43,44].

**Examination of bacterial persistence in kidneys and joints**. The kidneys of the
mice were aseptically removed and assessed by three investigators in a blinded
manner (Z.H., T.J., and M.M.). For abscesses, a scoring system range from 0–3
points was carried out as previously detailed[43]. Later, the kidneys were homo-
genized, diluted serially in PBS, and transferred to agar plates containing 5% horse
blood. The plates were incubated overnight at 37 °C and the bacterial numbers
were quantified as CFUs.

To assess the bacterial persistence in joints of mice, joints were collected into
individual Eppendorf tubes containing 1 ml of PBS, followed by homogenization by
TissueLyser II (Qiagen, Hilden, Germany). Homogenized samples were inoculated
on horse blood agar plates by inoculation loops (1 μl, SARSTEDT, Nümbrecht,
Germany). The plates were incubated at 37 °C overnight and quantified as CFUs.
Bacteria equal or more than 10 CFU/joint were considered as positive[28].

**Micro-computed tomography (μCT)**. Joints were fixed in 4% formaldehyde for a
period of 3 days and then transferred to PBS for 24 h. All 4 limbs were scanned using
a Skyscan1176 micro-CT (Bruker, Antwerp, Belgium) with a voxel size of 35 μm and
subsequently reconstructed into a three-dimensional (3D) structure. The scanning
was conducted at 55 kV/ 455 μA, with a 0.2-mm aluminum filter. The exposure time
was 47 ms. The X-ray projections were obtained at 0.7° intervals with a scanning
angular rotation of 180°. The projection images were reconstructed into 3D images
using NRECON software (version 1.5.1; Bruker) and analyzed using CT-vox (version
2.4; Bruker). Each joint was blindly evaluated by two individual observers (Z.H. and
T.J.) using a scoring system[1] from 0 to 3 (0: healthy joint; 1: mild bone destruction; 2:
moderate bone destruction; and 3: marked bone destruction).

**Measurement of cytokine and chemokine levels**. The levels of TNF-α, IL-6,
interferon-γ (IFN-γ), macrophage inflammatory protein 2 (MIP-2), KC in the
serum samples were determined using a DuoSet ELISA Kit (R&D Systems Europe,
Ltd) according to manufacturer's instructions.

**Flow cytometry**. Whole blood was collected into EDTA coated tube. For 1 ml
whole blood, 9 ml of eBioscience™ 1X RBC Lysis Buffer (Invitrogen, Waltham, MA,
USA) was added for red blood cells (RBCs) lysis, after 10 min, cells were cen-
trifuged and resuspended in flowcytometry buffer (3% Heat inactivated FCS, 1 mM
EDTA, 10 mM HEPES, 0.09% NA acid), 2 million cells were blocked with 2 μl
Mouse BD FcBlock™ (BD Biosciences) for 5 min on ice, cells were resuspended in
antibody cocktail for 20 min after centrifugation, then, washed twice with cold PBS
and stained with FITC Annexin V kit (Biolegend, San Diego, CA, USA) and
7-AAD (Invitrogen, Waltham, MA, USA) according to manufacturer's instruc-
tions. Antibodies used in the antibody cocktail can be found in Supplementary
Table 1. Samples were acquired using BD FACSLyric™ flow cytometry (BD
Biosciences). UltraComp eBeads™ Compensation Beads (Invitrogen, Waltham,
MA, USA) were used to set up the compensation. Fluorescence minus one (FMO)
samples were used to identify negative population for each antibody. Data analyze
were performed by using FlowJo software (version 10.8; Tree Star, Ashland, USA).

**Statistics and reproducibility**. Statistical analyses were performed using Graph-
Pad Prism version 9 (GraphPad Software, La Jolla California, USA). Comparison
between experimental groups was performed using the Mann-Whitney *U* test, the
Fisher's exact test, paired t test and the Mantel-Cox log-rank test as appropriate. All
results are reported as mean ± standard error of the mean unless otherwise spe-
cified and *P*-value < 0.05 was considered as statistically significant. Numbers of
repeats for each experiment are described in the associated figure legends.

**Reporting summary**. Further information on research design is available in the Nature
Research Reporting Summary linked to this article.

## Data availability
The authors declare that the main data supporting the findings of this study are available
within the article and its Supplementary files. Source data underlying plots shown in
figures are provided in Supplementary Data 1–8. Extra data are available from the
corresponding author upon request.

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

## Acknowledgements

This work was supported by the Swedish Medical Research Council (grant number 523-2013-2750 and 2019-01135 to T.J.); grants from the Swedish state under the agreement between the Swedish Government and the county councils, the ALF-agreement (grant number ALFGBG-823941, ALFGBG-933787 and ALFGBG-965074 to T.J., ALFGBG-770411 to A.J., ALFGBG-926621 to R.P.); E och K.G. Lennanders stipendiestiftelse to M.M.; Rune och Ulla Amlövs Stiftelse för Neurologisk och Reumatologisk Forskning to T.J, P.K.K. and M.M.; Sahlgrenska University Hospital Foundations to A.J. and M.M.; Stiftelsen Wilhelm och Martina Lundgrens Vetenskapsfond to P.K.K.; National Natural science foundation of China (Grant number 81460334) to Y.F.; The Innovative Talents Team Program of Guizhou Province (Grant number. 2019-5610) to Y.F.; and Gothenburg University. F.G. were funded by the Deutsche Forschungsgemeinschaft (DFG) Germany´s Excellence Strategy—EXC 2124—390838134 'Controlling Microbes to Fight Infections'. The funders had no role in study design, data collection and analysis, decision to publish, or preparation of the manuscript. We are grateful to Libera Lo Presti (Excellence Cluster 2124 'Controlling Microbes to Fight Infections' (CMFI), University of Tübingen, Germany) for critical reading of the manuscript.

## Author contributions

T.J., Z.H., H.F., C.D., Y.F., R.P., and F.G. conceived and planned the experiments. Z.H., P.K.K., P.E., M.M., S.L., A.J., M.S., M.D., M.N., and M.T.N. carried out the experiments. Z.H. and T.J. wrote the manuscript. All authors contributed to the interpretation of the results and provided critical feedback and helped shape the research, analysis and manuscript. All authors contributed to the article and approved the submitted version.

## Funding

## Competing interests
The authors declare no competing interests.
