## [Peer Review File · Communications Biology]

Reviewers' comments:

Reviewer #1 (Remarks to the Author):

This manuscript shows that PSM β reduced the severity of septic arthritis in mouse model caused by *S. aureus*. It may promote the researches in septic arthritis caused by bacteria. The evidences on the activation of TLR2, which is mentioned in the manuscript, should be provided. In the manuscript, it is also mentioned as " Δ PSM β strain caused similar mortality as its parental strain", and it might be contradicted to "PSM β expression protected from the development of septic arthritis." in conclusion. In this case, the discussion on this point should be more carefully and in details.

Reviewer #2 (Remarks to the Author):

The manuscript from Hu et al investigates the role of phenol soluble modulins in a murine model of septic arthritis. In vitro experiments were done to investigate the activation of neutrophils by PSM alpha and beta. These experiments showed that both PSMs have different roles: PSM alpha activates neutrophils but not PSMbeta. Furthermore, the activation of neutrophils induced by PSM alpha can be blocked by PSM beta.

The authors performed a murine model of septic arthritis to investigate the role of both PSMs in the development of infection. By using deficient PSM strains, the authors demonstrated that the lack of PSM beta increased the severity of arthritis in mice. This effect has been shown by the analysis of bone erosion with all the strains.

The manuscript is well-written and easy to follow. However, I have some questions/suggestions:

1. PSMbeta is a conditional neutrophil agonist: please explain the use of fMLF. It is explained in the mat and methods but not in this section
2. Fig. 2: I suggest replacing KRG for buffer or adding this information in the legend of fig. 2.
3. Fig. 3C: what is the n? Did you perform the statistic in this experiment? Is the reduction of O₂ production significantly reduced with doses of PSMB higher than 125nM?
4. What is the difference between WKYMVM and F2Pa110? Why do you think that PSMB induced different effects of both agonists? Please explain/discuss in your discussion.
5. Why the O₂ production induced by PSMB1 is different in fig.3A and fig. 1?
6. Did the authors investigate the CFU/ml in bones/joints?
7. I understand that the authors investigated the CFU/ml in kidneys after 10 days. Thus, the authors can speak about bacterial persistence and/or clearance. It seems that the lack of PSM alpha affects the persistence of *S. aureus* in kidneys...Please explain this point in your discussion
8. Why the cytokines were measured after 10 days in serum? Is it possible to measure after 1d of infection as well? It looks like the effect of the strain deficient of psmb is important in the first stage of infection.
9. The authors assessed the expression of adhesins in all strains. What about other virulence factors that may contribute to the development of sepsis/arthritis such as hla?
10. Please include in your discussion the role of PSMs during infection. Why *S.aureus* produce both PSMs? What is the advantage for this microorganism to produce 2 antagonist molecules during infection? Maybe PSM alpha is produced to colonize, spread to other tissues and escape from the immune system and psmb more for survival?

Reviewer #3 (Remarks to the Author):

Phenol-soluble modulins (PSMs) have emerged as critical virulence determinants of *S. aureus*. While PSM alphas have been associated with virulence and cytolysis, and the delta-toxin also with specific virulence-associated phenotypes, a specific role of the larger PSM betas has remained elusive. It is known that PSM betas are generally barely cytolytic to human cells and have reduced pro-inflammatory activity as compared to other PSMs. In *S. aureus*, they are only produced in small amounts.

Main findings of the present report are that PSM beta peptides reduce the pro-inflammatory activity of PSMalpha3 on human neutrophils as measured by the induction of superoxide and a

psm beta ko strain in the Newman background causes higher virulence in a septic arthritis model.

While the inhibitory effect of PSM betas on PSMalpha3-mediated neutrophil activation is certainly interesting, the manuscript in large parts reproduces what is already known and fails to provide other than somewhat superficial characterization of the underlying mechanisms. Main shortcomings that the limit significance of this report include:

- It has been shown previously, in the initial 2007 Wang et al. paper on *S. aureus* PSMs, that a psm beta ko in the clinical strain LAC is more virulent in a skin infection model, but not in a systemic infection (sepsis) model. The present findings in an arthritis model are interesting, but this is therefore not the first time that a higher-virulence phenotype of a psm beta ko in *S. aureus* infection has been demonstrated. Corresponding sepsis results have also already been shown in the Wang paper.
- The mechanism of PSM beta interference with PSMa3-mediated neutrophil activation remains unclear.
- PSM ko strains are not used in the in-vitro neutrophil experiments. Given that (i) PSM betas are only produced in low amounts in *S. aureus* and (ii) previous results have shown that a psm beta ko does not impact neutrophil activation as measured by calcium flux (Kretschmer et al. *Cell Host Microbe* 2010), it remains unclear whether the results obtained here with synthetic peptides have any biological relevance, particularly as the in-vivo experiments do not link to neutrophil activation.
- Related to linking in-vivo phenotypes to neutrophil interaction, the authors performed some experiments meant to rule out other underlying reasons. However, there are several problems here as well. They performed qRT-PCR analyses of two surface proteins, but (i) PSMs may well have effects that are post-transcriptional such as demonstrated for their interaction with lipoproteins, and (ii) the impact of PSM presence on the expression of other genes has also already been analyzed in detail using microarrays. Furthermore, there seems to be a strong effect on PSM alpha gene expression and presence in the culture filtrate early during growth, which the authors completely glance over.
- *S. aureus* strains produce all PSMs, except for in quorum cheater (*agr*) mutants, which are totally devoid of PSMs. Only psm beta-deficient strains are not known to appear naturally or have clinical relevance.

Given lacking general novelty of the in-vivo findings and lack of clinical significance of PSM beta-deficient isolates, one would expect a much more detailed investigation of the mechanism underlying the impact of PSMs on neutrophil activation in vitro and a better demonstration that it is neutrophil interaction that affects the in-vivo differences in virulence regarding presence of PSM beta peptides. However, it remains quite unclear based on the data presented in this study what is really going on.

Gothenburg, July 12th, 2022

We would like to thank you and the referees for the insightful examination of our manuscript. In order to strengthen our conclusions, one *in vivo* experiment and several immunological and bacteriological assays were performed as reviewers advised including:

- 1) To study the immunological mechanisms at the early time in mice with *S. aureus* infections, NMRI mice infected with PSM mutants and their parental strain were sacrificed on day 3 after infection. The cytokine levels were measured by ELISA.
- 2) To find the connection between *in vitro* findings and *in vivo* data, we have analyzed the cell apoptosis and death of neutrophils in mice infected with those 3 strains on day 3 after infection.
- 3) The kidney cfu counts on day 3 were included.
- 4) The data regarding bacterial counts in joints were included.
- 5) Hemolysin assay by blood agar plates were used to understand whether depletion of *psms* affects the hemolysin expression in *S. aureus*.

Reviewers' comments:

Reviewer #1 (Remarks to the Author):

This manuscript shows that PSM β reduced the severity of septic arthritis in mouse model caused by *S. aureus*. It may promote the researches in septic arthritis caused by bacteria. The evidences on the activation of TLR2, which is mentioned in the manuscript, should be provided.

*: Now we have provided the cytokine data on day 3 after infection. We found that both IL-6 and KC were upregulated in Δ PSM β strain infected mice compared to WT and PSM α mutant infected mice. In the Gram positive bacteria (*S. aureus*) infection, the inflammatory responses are largely dependent on TLR2. The upregulation of IL-6 and KC may suggest the activation of TLR2.*

In the manuscript, it is also mentioned as " Δ PSM β strain caused similar mortality as its parental strain", and it might be contradicted to "PSM β expression protected from the development of septic arthritis." in conclusion. In this case, the discussion on this point should be more carefully and in details.

: Thank you for constructive comments. We have now described our findings regarding the mortality and septic arthritis more in details.

Reviewer #2 (Remarks to the Author):

The manuscript from Hu et al investigates the role of phenol soluble modulins in a murine model of septic arthritis. *In vitro* experiments were done to investigate the activation of neutrophils by PSM α and β . These experiments showed that both PSMs have different roles: PSM α activates neutrophils but not PSM β . Furthermore, the activation of neutrophils induced by PSM α can be blocked by PSM β .

The authors performed a murine model of septic arthritis to investigate the role of both PSMs

in the development of infection. By using deficient PSM strains, the authors demonstrated that the lack of PSM beta increased the severity of arthritis in mice. This effect has been shown by the analysis of bone erosion with all the strains.

The manuscript is well-written and easy to follow. However, I have some questions/suggestions:

1. PSMbeta is a conditional neutrophil agonist: please explain the use of fMLF. It is explained in the mat and methods but not in this section

: *Very good suggestion. A short sentence explaining the use of fMLF is now included in this section.*

2. Fig. 2: I suggest replacing KRG for buffer or adding this information in the legend of fig. 2.

: *We fully agree with you that 'buffer' is a better alternative.*

3. Fig. 3C: what is the n? Did you perform the statistic in this experiment? Is the reduction of O₂ production significantly reduced with doses of PSMb higher than 125nM?

: *The experiment was performed 3 times with different buffy coats. All experiments showed the similar pattern. The figure 2C is a representative figure. This information is now included in the figure legends.*

4. What is the difference between WKYMVM and F2Pa110? Why do you think that PSMb induced different effects of both agonists? Please explain/discuss in your discussion.

: *Thanks for the constructive comments. We have been thinking about this. Now we have briefly discussed this issue in the results.*

5. Why the O₂ production induced by PSMb1 is different in fig.3A and fig. 1?

: *O₂ production induced by PSMb1 was same in fig 1 and fig 3A. We are very sorry for the unclear figures causing confusion. PSMb1 did not cause any O₂ production when the neutrophils were not primed with TNF and the line was overlapped with x-axis in fig 3A. We have now changed the scales of fig 3 and the line colors to make the figures more clear.*

6. Did the authors investigate the CFU/ml in bones/joints?

: *Very good suggestion. Now the data about cfu in joints are included in the results part (Fig.6C).*

7. I understand that the authors investigated the CFU/ml in kidneys after 10 days. Thus, the authors can speak about bacterial persistence and/or clearance. It seems that the lack of PSM alpha affects the persistence of S. aureus in kidneys...Please explain this point in your discussion

: *PSMa has cytotoxic effect on neutrophils. It is therefore not so surprising that lack of PSMa affects the persistence of bacteria in kidneys. We have now explained this point in the discussion more clearly.*

8. Why the cytokines were measured after 10 days in serum? Is it possible to measure after 1d of infection as well? It looks like the effect of the strain deficient of psmb is important in the first stage of infection.

: *Thank you very much for very constructive comments, which indeed improved the quality of the study. Now we have measured the cytokine levels at the early time points of infection (day 3). The data are now presented in fig 7 A and B. Indeed, at the early phase of infection*

Δpsmβ infected mice had significantly higher levels of IL-6 (proinflammatory cytokine) and KC (neutrophil attractant) than WT strain infected mice, suggesting *Δpsmβ* infected mice had more severe systemic inflammation than WT infected mice.

9. The authors assessed the expression of adhesins in all strains. What about other virulence factors that may contribute to the development of sepsis/arthritis such as hla?

: Obviously, it is not practically feasible to assess the expression of all virulence factors in those 3 strains. Therefore, we chose some of the most well-known virulence factors in septic arthritis. Hla is a very important virulence factor and worth to be tested. We used the traditional blood agar plate hemolysin assay (semiquantitative methods) and found that the hemolysis was similar in those 3 strains. The data are now included in the supplementary materials (Sfig4).

10. Please include in your discussion the role of PSMs during infection. Why *S.aureus* produce both PSMs? What is the advantage for this microorganism to produce 2 antagonist molecules during infection? Maybe PSM alpha is produced to colonize, spread to other tissues and escape from the immune system and psmb more for survival?

: Interesting point. We have now discussed this issue in the discussion.

Reviewer #3 (Remarks to the Author):

Phenol-soluble modulins (PSMs) have emerged as critical virulence determinants of *S. aureus*. While PSM alphas have been associated with virulence and cytolysis, and the delta-toxin also with specific virulence-associated phenotypes, a specific role of the larger PSM betas has remained elusive. It is known that PSM betas are generally barely cytolytic to human cells and have reduced pro-inflammatory activity as compared to other PSMs. In *S. aureus*, they are only produced in small amounts.

Main findings of the present report are that PSM beta peptides reduce the pro-inflammatory activity of PSMalpha3 on human neutrophils as measured by the induction of superoxide and a psm beta ko strain in the Newman background causes higher virulence in a septic arthritis model.

While the inhibitory effect of PSM betas on PSMalpha3-mediated neutrophil activation is certainly interesting, the manuscript in large parts reproduces what is already known and fails to provide other than somewhat superficial characterization of the underlying mechanisms. Main shortcomings that the limit significance of this report include:

- It has been shown previously, in the initial 2007 Wang et al. paper on *S. aureus* PSMs, that a psm beta ko in the clinical strain LAC is more virulent in a skin infection model, but not in a systemic infection (sepsis) model. The present findings in an arthritis model are interesting, but this is therefore not the first time that a higher-virulence phenotype of a psm beta ko in *S. aureus* infection has been demonstrated. Corresponding sepsis results have also already been shown in the Wang paper.

: We fully agree with you that hypervirulence of PSMb deficient strain was also found before in skin infection models. We have now mentioned and cited the publication in the discussion. At the same time, septic arthritis and skin infections are two totally different diseases with distinct pathogenesis. Also, septic arthritis is more severe and aggressive disease than skin infections.

- The mechanism of PSM beta interference with PSMA3-mediated neutrophil activation remains unclear.

: Now we have provided substantial new data regarding the in vivo assessment of neutrophils and cytokine levels at the early time point of infection. Our data demonstrated that at the early phase of infection Δ psm β infected mice had significantly higher levels of IL-6 (proinflammatory cytokine) and KC (neutrophil attractant) than WT strain infected mice, suggesting Δ psm β infected mice had more severe systemic inflammation than WT infected mice. Indeed, we have shown before that IL-6 levels positively correlated to the severity of bone erosions in septic arthritis. Moreover, by FACS analyses, we found that Δ psm β infected mice tended to have more dead neutrophils than WT infected mice in blood (Both Δ psm β and WT infected mice had more dead neutrophils than psm α infected mice). Our data indicate that psm β may protect the neutrophils from cytotoxic effects of psm α . We believe that our new data have provided some mechanism for our in vivo observations.

- PSM ko strains are not used in the in-vitro neutrophil experiments. Given that (i) PSM betas are only produced in low amounts in *S. aureus* and (ii) previous results have shown that a psm beta ko does not impact neutrophil activation as measured by calcium flux (Kretschmer et al. Cell Host Microbe 2010), it remains unclear whether the results obtained here with synthetic peptides have any biological relevance, particularly as the in-vivo experiments do not link to neutrophil activation.

: We have now included the data of neutrophil apoptosis and death in vivo on day 3 after infection. It is clear that the percentage of neutrophils in leukocytes were significantly lower in both WT and Δ psm β infected mice compared to Δ psm α infected mice. Importantly, more dead neutrophils were found in both WT and Δ psm β infected mice compared to Δ psm α infected mice. There was a trend that Δ psm β infected mice had more dead neutrophils than WT infected mice. Our data may indicate that psm β somehow protects the neutrophils from cytotoxic effects of psm α .

- Related to linking in-vivo phenotypes to neutrophil interaction, the authors performed some experiments meant to rule out other underlying reasons. However, there are several problems here as well. They performed qRT-PCR analyses of two surface proteins, but (i) PSMs may well have effects that are post-transcriptional such as demonstrated for their interaction with lipoproteins, and (ii) the impact of PSM presence on the expression of other genes has also already been analyzed in detail using microarrays. Furthermore, there seems to be a strong effect on PSM alpha gene expression and presence in the culture filtrate early during growth, which the authors completely glance over.

: We agree with all points that you addressed. All the issues that you have addressed are discussed in the discussion now.

- *S. aureus* strains produce all PSMs, except for in quorum cheater (*agr*) mutants, which are totally devoid of PSMs. Only psm beta-deficient strains are not known to appear naturally or have clinical relevance.

: We agree with you that psm beta deficient strain do not appear naturally. However, the use of psm deficient strain is to understand the mechanism rather than exploration of clinical relevance.

Given lacking general novelty of the in-vivo findings and lack of clinical significance of PSM beta-deficient isolates, one would expect a much more detailed investigation of the

mechanism underlying the impact of PSMs on neutrophil activation in vitro and a better demonstration that it is neutrophil interaction that affects the in-vivo differences in virulence regarding presence of PSM beta peptides. However, it remains quite unclear based on the data presented in this study what is really going on.

: There are technical difficulties to study the mechanism underlying the impact of PSMs on neutrophil activation in vitro by using the live bacteria. Therefore, we designed the experiment to study the neutrophil apoptosis and death at the early phase of infection (Day 3 post infection). Now we have provided substantial new data regarding the in vivo assessment of neutrophils and cytokine levels at the early time point of infection. Our data demonstrated that at the early phase of infection $\Delta psm\beta$ infected mice had significantly higher levels of IL-6 (proinflammatory cytokine) and KC (neutrophil attractant) than WT strain infected mice, suggesting $\Delta psm\beta$ infected mice had more severe systemic inflammation than WT infected mice. Moreover, by FACS analyses, we found that $\Delta psm\beta$ infected mice tended to have more dead neutrophils than WT infected mice in blood (Both $\Delta psm\beta$ and WT infected mice had more dead neutrophils than $psm\alpha$ infected mice). Our data indicate that $psm\beta$ may protect the neutrophils from cytotoxic effects of $psm\alpha$. We believe that our new data have provided some mechanism for our in vivo observations.

We hope that our improved manuscript will be now acceptable for publication in your Journal.

Yours sincerely,
Tao Jin, MD, PhD, Associate professor

REVIEWERS' COMMENTS:

Reviewer #2 (Remarks to the Author):

The authors answered all my questions and improved a lot the manuscript. I am happy to accept the manuscript.

Reviewer #3 (Remarks to the Author):

I would have liked to see more detailed investigation of the underlying mechanism but I understand that this may be the subject of follow-up studies. I appreciate the experiments and text changes that were added and am happy with the manuscript as it stands.